# Modelling and Analysis of Hybrid Transformation for Lossless Big Medical Image Compression

**DOI:** 10.3390/bioengineering10030333

**Published:** 2023-03-06

**Authors:** Xingsi Xue, Raja Marappan, Sekar Kidambi Raju, Rangarajan Raghavan, Rengasri Rajan, Osamah Ibrahim Khalaf, Ghaida Muttashar Abdulsahib

**Affiliations:** 1Fujian Provincial Key Laboratory of Big Data Mining and Applications, Fujian University of Technology, Fuzhou 350011, China; 2School of Computing, SASTRA Deemed University, Thanjavur 613401, India; 3Department of Solar, Al-Nahrain Renewable Energy Research Center, Al-Nahrain University, Baghdad 64040, Iraq; 4Department of Computer Engineering, University of Technology, Baghdad 19006, Iraq

**Keywords:** big data, data security, knight tour, steganography, wavelet transform, lossless compression

## Abstract

Due to rapidly developing technology and new research innovations, privacy and data preservation are paramount, especially in the healthcare industry. At the same time, the storage of large volumes of data in medical records should be minimized. Recently, several types of research on lossless medically significant data compression and various steganography methods have been conducted. This research develops a hybrid approach with advanced steganography, wavelet transform (WT), and lossless compression to ensure privacy and storage. This research focuses on preserving patient data through enhanced security and optimized storage of large data images that allow a pharmacologist to store twice as much information in the same storage space in an extensive data repository. Safe storage, fast image service, and minimum computing power are the main objectives of this research. This work uses a fast and smooth knight tour (KT) algorithm to embed patient data into medical images and a discrete WT (DWT) to protect shield images. In addition, lossless packet compression is used to minimize memory footprints and maximize memory efficiency. JPEG formats’ compression ratio percentages are slightly higher than those of PNG formats. When image size increases, that is, for high-resolution images, the compression ratio lies between 7% and 7.5%, and the compression percentage lies between 30% and 37%. The proposed model increases the expected compression ratio and percentage compared to other models. The average compression ratio lies between 7.8% and 8.6%, and the expected compression ratio lies between 35% and 60%. Compared to state-of-the-art methods, this research results in greater data security without compromising image quality. Reducing images makes them easier to process and allows many images to be saved in archives.

## 1. Introduction

In recent years, there has been a need for extensive medical data information security in all research fields. One such field where information security is crucial in medicine and pharmacology. Today, the number of diseases and people affected by them has increased, posing a potential challenge in storing various medically significant data in limited storage spaces. In addition, data security cannot be compromised. Over the years, steganography techniques have proven to be one set of effective methods to maintain the privacy of big medical data. This method uses an overlay image to embed and retrieve data algorithmically. There are several options for entering information into big medical data. One such variation used in this paper is the knight tour (KT) traversal technique combined with least significant bit (LSB) substitution to preserve data. The big data of the patient are added to the medical image, which is the cover image of this steganography technique. The KT performs the embedding, and the binary information is embedded in the LSB position of the medical big data image pixels placed in the KT path. To increase security, the discrete wavelet transform (DWT) is applied to the cover image to make it unreadable, and lossless wavelet compression is used to efficiently store the transformed image in an extensive medical data repository. This hybrid approach ensures the security of patient information and the efficient storage of information in a medical database using as little storage space as possible.

Initial attempts to implement communications, photo archiving systems and teleradiology applications have encountered significant obstacles. Large picture files present problems for both transmission and storage due to the high cost of the equipment required to manage them. Lossless compression techniques only accomplish 2:1 to 3:1 reductions for medical photos by using redundancy within an image to transfer image information more effectively while permitting flawless reconstruction. Images can be reduced by arbitrarily huge ratios using irreversible or “lossy” processes, but the original pictures are not fully replicated. However, the reproduction quality could be sufficient to prevent visible picture deterioration and diminished diagnostic usefulness. This paper examines wavelet compression in a new light. We compare the method to the famous Joint Photographic Experts Group (JPEG) approach. We review methods for measuring compression algorithms’ effectiveness and look at new developments in wavelet compression.

The motivations behind this research work, along with the significant contributions respecting processes, are as follows:▪Compressing the image—Apply wavelet analysis to separate and approximate the image information based on the sub-images and sub-signals.▪Pixel approximation for compression—Apply the images’ horizontal, vertical, and diagonal features and approximate the sub-signals for better pixel approximation.▪Frequency and time localization—Apply WT effectively to reduce computational complexity.▪Signal-features-based compression—Apply WT to distinguish between the minute signal features during the compression.▪Quantization and lossless compression—Apply the hybrid transformation to quantize the coefficients to optimize the lossless image compression during the transmission.

Wavelet compression and JPEG fall under the “transform-based lossy compression algorithms” umbrella category. Transformation, quantization, and encoding are the three processes that make up these approaches. A picture is converted from grayscale values in the spatial domain to coefficients in another domain in a lossless procedure known as transformation. The Fourier transform, used to recreate magnetic resonance (MR) pictures, is one well-known transform. The DCT and WT are two more transformations more often employed in compression. The transformation phase does not result in any information loss. Information is lost during the quantization process. While less significant coefficients are loosely approximated, frequently as zero, efforts are made to preserve the more significant coefficients. Converting floating points to decimals might be enough to perform quantization: values are converted to integers. The quantized coefficients are finally encoded. The quantized coefficients are compactly represented in this step, which is also lossless, to allow for adequate image storage or transmission.

The paper is organized as follows. Section 2 presents a literature survey, Section 3 discusses the proposed model, and Section 4 focuses on the results and analysis. The conclusions and a consideration of future enhancements are drawn in Section 5.

## 2. Literature Survey and Critiques

Various steganography methods and standard guidelines have recently been proposed that provide valuable recommendations and suggest using object-oriented mechanisms [1]. A three-layer technique, including pixel statistics preservation, Moore space-filling curves, and Hilbert space-filling curves, increases and improves security [2]. A detailed comparative analysis was made of different digital steganography techniques, and performance was evaluated in terms of peak signal-to-noise ratio (PSNR) and mean square error (MSE) values [3]. Image compression of black and white images using a wavelet transform that uses the LGB algorithm and an error-correction method to minimize distortion has been presented. The compression result was compared with other techniques for encoding DWT-coupled map grid (CML) data [4]. The results show that the proposed encryption algorithm has the advantage of large key space, high security, and fast encryption or decryption speed [5]. A recently developed generic codec supporting JPEG 2000 with volumetric extension (JP3D) was used to investigate how to optimally compress volumetric medical images using JP3D. Various directional WTs and a general in-group prediction mode provide guidelines and settings for optimal compression of medical volumetric images at an acceptable level of complexity [6].

A low-complexity 2D image compression method was tested using Haar waves, and compressed image quality was obtained. Some factors, such as compression ratio (CR), PSNR, mean opinion score (MOS), picture quality scale (PQS), etc., have been used to evaluate the quality of compressed images [7]. The work used a simple LSB substitution technique for data hiding. The MSE is calculated, and the resulting image has no significant changes from the original image [8]. An advanced steganography technique that includes side information to calculate the correlations between neighboring pixels and estimate degrees of smoothness has been used. The results showed that the technology provides a high delivery capacity with less distortion [9]. Steganography was used by adjusting the LSB value of the color pixel intensity by simple binary addition. The results showed that the embedding power was twice that of the traditional technique [10]. The methods used were DWT for medical image compression. Research shows that correlation and redundancy are reduced in the DWT domain, while random permutation of pixels with an encryption key leads to confusion and dispersion [11]. Information hiding was developed with multi-pixel difference (MPD) and LSB (least significant bit) replacement to improve image quality and increase information storage capacity. The results showed that the improved image and resolution were undistorted [12]. Concatenative singular value decomposition (SVD) and optimized DWT were used. Huffman encoding and decoding were also performed.

The results were compared with other image compression methods based on CR, PSNR, SSIM, and MSE [13]. The various scenography and digital watermarking techniques for beginners were explored and acted as a guide to help understand the concepts and apply them very easily [14]. The coefficients of the applied 2-D orthogonal WT and the transformed image were coded and quantified according to the local estimated noise sensitivity. The human visual system offers a high degree of compression [15]. In image compression, fusion, and encryption, big medical data CS, chaos, and fractional Fourier transforms were used simultaneously. The technique was developed to reduce data and simplify the keys [16]. Secure and lossless digital image watermarking was developed based on DWT and DCT databases to preserve patient privacy in a medical database. Performance development factors include PSNR measurements and their correlations with overall image degradation [17]. DWT was used for the fast and secure transmission of primary images online. The results showed that the algorithms quickly and securely send essential images in a group [18]. A wavelet-based approach was developed to compress and encrypt fused images by selecting salient and less salient information. Fusion is performed by error measurement with compression, and encryption methods use pseudo-random number sequences and Huffman coding. The results showed that the proposed method was better than all others [19]. DWT was used for medical big data image compression. The results showed that the proposed algorithm has better security and performance than previous works [20]. This research used DCT, DWT, and watermark methods to store patient data. The results showed that the proposed system performs better un-detectability [21]. The modulation function was applied, and many advantages were achieved regarding good resolution, overcoming LSB compensation, and creating large storage space to achieve a sizeable hidden volume [22]. The Embedded Zero tree Wavelet (EZW) algorithm provides an efficient technique for low bit rates and high compression and advanced EZW techniques for lossless and lossless grayscale and color image compression [23]. The simple and optimized LSB replacement method and genetic algorithm were developed for image embedding. The results showed that the embedded image was not significantly affected, and the hiding strategy was improved [24]. The new type of steganography using the LSB replacement and pixel value difference (PVD) methods was developed to improve image quality and maximize the hidden space [25,26,27,28].

The entropy, run length, and dictionary-based compression techniques were developed to achieve lossless compression [29,30,31,32,33]. The DNN and principal component analysis strategies were applied to predict data compression using entropy values. The images were divided into blocks, and then applied compression [34].

## 3. Proposed Methodology

The proposed method mainly focuses on KT steganography, WTs, and compression. Today, image and audio file sizes are drastically compressed in a new mathematical way. Lossless compression is an old technology, but JPEG and MP3 now use lossless compression and are essentially required in the new environment. The original document was extensively searched using a mathematical model. Typically, compression up to a tenth of the original image length is possible. The quality and accuracy never deteriorate. Record compression and image compression must not allow images to be lost. The length of the original image must match the load image after it is translated.

The fast KT approach is designed to move only in the L direction, i.e., three down and one to the right, starting at the first pixel of the image. The cover image is converted using DWT, and the “wpdencmp” lossless wavelet packet based on the “Haar” wavelet is used to compress the transformed image. The complete methodology is described in the architecture diagram (Figure 1). DWT has wide applications in technological understanding, engineering, mathematics, and computer engineering knowledge. In particular, it encodes characters to symbolize separate records in an extra-redundant format, often as a prerequisite for fast compression. Both strategies have their advantages and disadvantages. Like DWT, it offers a higher compression ratio of 1:3 without losing photos but still requires more processing power. DCT can perform electrical processing, and artifact blocks are like losing some statistics’ continuous WTs.

### 3.1. Architecture Diagram

Image compression is a method used to reduce image space. In decompression, a small part of the image is first removed and scaled to the large image. The primary purpose of image compression is to reduce minor images and noise because images are removed. Image redundancy is squared to make the image effective. This technique reduces the bit size of an image without affecting image quality. Reducing the image quality does not affect the image quality of any account. The shorter the length, the more images can be stored under the archive. The disk space required to store images is small or large.

### 3.2. Proposed Algorithm

The clinical data repository combines statistics from various clinical resources, including EMR or laboratory systems, to provide a complete picture of a patient’s care. These archives are characterized as databases containing scientific statistics. Healthcare extensive records is a term used to describe the large number of records created by adopting digital technologies that collect patient statistics and help manage healthcare services that are otherwise too large and complex for classic technologies. Clinical repositories can provide a rich overview of patients, their clinical conditions, and outcomes. The database can provide a way to explore associations and capacity styles between disease progression and treatment. The proposed algorithm with improved embedding is sketched in Algorithm 1.

The DWT coefficients were first tested (approximate and detail coefficients are processed separately) by inserting zeros between each coefficient, doubling each length exactly. Just as the concept of filter financial institutions can determine DWT, so can IDWT reconstruction be performed by taking the real first N/2-1 coefficients of the DWT coefficients and adding them to the transfer. RMSE suggests that square deviation is one of the most typically used measures for comparing excellent predictions. It shows how some distance predictions fall from measured actual values using Euclidean distance. To compute RMSE, calculate the residual (the distinction between prediction and fact) for each statistical factor, compute the residual norm for every information factor, compute the suggestion of residuals, and take the square root of that mean. RMSE is typically utilized in supervised mastering applications, as RMSE uses and requires accurate measurements for each predicted record factor.
**Algorithm 1:** Improved Embedding1: Load the medical image and the patient data as X ← medical image and D ← patient data.2: Adjust the matrix dimensions such that the rows are multiples of 3 and the columns of 2.3: Pad the extra row or column with the value ‘256′ as the pixel ranges between 0 and 255.4: Extract the ASCII value of D and convert it to its equivalent binary value as D → ASCII (D) → Binary(D).5: Apply KT for pixel traversing the image in the ‘L’ path, and change the LSB bit of each pixel with each bit of the binary value from D.6: Delete the extra padded row or column.7: Extract the red, green, and blue components from X.8: Apply forward DWT for the red, green, and blue components using the formula
(1)$$I_{DWT}\left(a,b\right)=2^{-a/2}\int_{-\infty }^{+\infty }i\left(t\right)\psi \left(2^{-a}t-b\right)dt$$
(2)$$\psi_{a,b}\left(t\right)=2^{-a/2}\psi \left(2^{-a}t-b\right)$$
where ψ(t) is the mother wavelet.9: Combine the individual transformed components into one matrix.trans = combine (red, blue, and green)10: Compute lossless wavelet packet compression using ‘wpdencmp’ function with ‘haar’ wavelet packet and store it in the repository.Z = compress (wpdencmp, trans, haar)

The extraction algorithm is defined in Algorithm 2 as follows.
**Algorithm 2:** Extraction1: Retrieve the compressed image from the medical database and perform wavelet decompression.trans = decompress (Z)2: Apply Inverse DWT (IDWT).(3)it=∑a∑bIDWTa,b2−a/2ψ2−at−b3: Extract the red, blue, and green components from the reconstructed image, and combine them.4: Make the image matrix traversable by adding an extra row or column and pad with the value ‘256′.5: Apply the KT to extract the LSB values from the pixels of the image traversing in an ‘L’ pattern.6: Delete the extra padded row or column.7: Convert the obtained binary bits to ASCII values and the character.Binary (D) = ASCII (D) = char (D) = D8: Update the patient data = D and medical image = X.

The proposed image compression is defined in Algorithm 3.
**Algorithm 3:** Proposed Image Compression1: Apply the improved embedding to the medical image.2: Perform the extraction operation.3: Apply the hard threshold using            yhardt=xt when absxt>T             yhardt=0,    when absxt≤T 4: Perform the soft threshold using            ysoftt=xt−T when absxt>T             ysoftt=0,    when absxt≤T 5: Find the entropy encoding h(s).            hs=−∑j=1qpsilog psi 6: Determine the Shannon entropy M(C(b))).7: Apply the wavelets based multi resolution analysis.8: Define the Haar transformIk,  n=[k2−n, k+12−n]9: Perform the dilation using             ∅x=2∑k∈z∅−k+2xhk10: Apply the wavelet operation using            φx=2∑k∈zg−k+2xgk11: Perform filtering operations using orthogonal quadrature.            hi=∑j=−∞∞uj.h−j+2i            gi=∑j=−∞∞uj.g−j+2i12: Apply the filters using bi-orthogonal quadrature.13: Decompose the image using wavelets and apply the color conversion.            YCbCr=0.30.60.1−0.2−0.30.50.5−0.4−0.1RGB14: Find RGB using             RGB=10.11.310.3−0.511.8−0.1YCbCr15: Apply the lossless compression through            YrVrUr=2G+R+B/4R−GB−G            GRBr=Yr−Vr+Ur/4G+VrG+Ur

### 3.3. Computational Performance of Wavelet Transform

The size of the filter set and the number of pixels in the image impact how quickly wavelet compression works. Although JPEG speed is related to the number of pixels, the DWT has substantially higher memory needs since the entire image must be changed simultaneously. In light of this, wavelet compression speed on a particular platform will be equivalent to JPEG compression speed for small pictures that can be stored entirely in memory but slower for bigger images. As an electronic file, the number of radiologic tests varies little between treatment techniques. A computed tomographic (CT) picture is 189 MB in size. A study with 40 images takes up roughly 16 MB for a two-view radiographic study and 20 MB for a study with 40 images. A typical MR picture is 8 MB; however, because many images are often collected, MR imaging examinations typically consume 10–15 MB. A cross-sectional picture may be decompressed using Windows NT 4.0 software (Microsoft, Redmond, Wash) in nanoseconds on a 200 MHz Pentium Pro computer with 64 MB of random access memory. A 10 MB radiograph may be decompressed using wavelet compression in around 9 s, compared to 7 s using the JPEG method. Standard modems may often operate via telephone lines at 28.8 kbit/sec rates. A 15 MB file would take roughly 90 min to transfer. A 10:1 compression ratio would cut the transmission time to 9 min, and a 33:1 ratio would lower it to 3 min. The improvement in transmission time is directly proportionate to the compression ratio employed. The compression ratio also linearly reduces the storage needed; for example, a 15:1 compression ratio would reduce the 15 MB research to just 1 MB [25,26,27,28,29,30,31,32].

## 4. Simulation and Analysis

The overall methodology of this research includes KT steganography, DWTs, and lossless wavelet packet compression of medical images for privacy, preservation, and efficient storage. The steganography technique uses the LSB replacement technique, which replaces LSB bits of pixels with patient data in binary form.

### 4.1. Datasets

The proposed model was simulated using datasets from http://www.aylward.org/notes/open-access-medical-image-repositories, accessed on 1 January 2023 [20,21,22,23,24]. The proposed model was tested using 43,752 chest radiographs from the National Institutes of Health Clinical Center (NIHCC) database. This comprises 1,00,000 images with relevant details, diagnostic information, and other publicly available datasets. The model was tested for various image sizes. The expected results were obtained with various data fixes and changes, which challenged the embedding algorithm, and additional data were included to verify the effectiveness of the compression algorithm.

### 4.2. Results and Analysis

The results of the steganography technique are shown in Figure 2; the changes in both the cover image and the steganography image cannot be distinguished. This demonstrates the superiority of the LSB replacement method in the field of steganography. The resulting steganography image is then transformed to render the image unreadable, and compression techniques are used to store the medical image in a database properly. This scenario is illustrated in Figure 3. This diagram shows the order of conversion, compression, decompression, and inverse conversion processes. To realize this methodology, we took as input cerebral hemorrhage steganography images already embedded in patient data and applied the DWT to obtain the transformed images.

The “dwt2” method is used for the conversion. In the DWT process, the original image is decomposed into up to two levels using the “Haar” wavelet. This decomposition produces horizontal, vertical, diagonal, and proximity components. The decomposed components are reconstructed using the DIWT to recover the original image. This strategy is illustrated in Figure 2, and the compression method takes a decomposed image as input and compresses it using a wavelet packet compression technique using “Haar” wavelet packets. The compression method, “wpdencmp,” uses a soft thresholding technique that uses wavelet packets to compress the image and compute the threshold. This compression concept is illustrated in Figure 3, along with histograms of the original and compressed images. A step-by-step compaction process is shown in Figure 4. At each level, the images are refined, and the differences between levels are visible. The higher the number of coding levels of compression, the higher the image’s compression ratio and recovered energy. The compressed image is decompressed by wavelet packet reconstruction using the accounting matrix values of the decomposed image. Horizontal, vertical, diagonal, and proximity components are extracted from the decompressed image, and an IDWT is applied to recover the original image. A title image embedded in the patient data is visualized.

In Figure 3, the steganography image of the brain hemorrhage is taken, and DWT is applied. After compression, the compressed image is displayed as a bar graph. The wavelet reconstruction method recovers the compressed image from the compressed image. The target image is retrieved from the decompressed image by applying the IDWT method to the previous step. Figure 4 represents the DWT and IDWT of the brain hemorrhage image. A sample segmentation of the transformed image is also displayed.

Figure 5 represents the histograms of the steganography and decomposed and compressed images in red, blue, and pink colors, respectively. The obtained outcome shows that the adopted methodology is efficient regarding high-resolution lossless compression of medical images. In the resultant images, the novelty of the work is apparent; it can be seen that the proposed methodology enables excellent elaboration and enhancement using improved WTs and lossless compressions. Figure 6 represents the transition steps of the wavelet packet compression technique, where the images are clearly distinguished from one another.

As shown in Figure 7, the complete process of wavelet photograph compresion is carried out and explored as follows. The laptop takes an entered image, before which wavelet redecoration has been finished for the digital image, thresholding has been executed on the digital picture as shown in Figure 8, and entropy coding has been completed for the photograph, all of which are essential. Wavelet compression is a technique that lets one minimize the dimensions of files at the same time as the equal time, improving them via the elimination of high-frequency noise additives. The documents can, without problems, be decreased beneath 1% of their actual length. Wavelet radically change, used to look up a sign into high-quality frequency factors at incredible choice scales, allows a picture’s spatial and frequency attributes to be concurrently revealed. In addition, competencies that could be undetected at one decision point might be easy to spot at another. One of the most vital blessings of wavelets is that they provide simultaneous localization in time and frequency areas. The most requirement in acquiring wavelets is that fast wavelet remodeling is computationally very speedy. Wavelets have the great advantage of being successful in isolating sign information. Lossless compression is more significant for programs where maintaining extraordinary files is essential. The drawback of this compression method is that it calls for massive archives to hold files after compression.

### 4.3. Comparative Analysis

The decompressed images of different sizes were compared with the original image. Several image measures—mean square error (MSE), signal-to-noise ratio (SNR), peak signal-to-noise ratio (PSNR), structural similarity index measure (SSIM), and bit error rate (BER)—were evaluated. The results are shown in Table 1. The performance of the proposed model was compared with those of other models—steganography, WT, segmentation, encryption and fusion, DCT, DWT, and classification [1,2,3,4,8,9,14,15,16,17,19,21,23,24].

This research proves that the evaluation parameters do not change significantly as the size of the information increases and gives the insight that, regardless of size, the error rate is low enough that the steganography or decompressed image will not be distorted. Regardless of changes to the input data, the embedding and compression algorithms do not deviate from expected results, proving robustness and accuracy to any dataset and the tricky challenges introduced. The compression ratios of all images also exceeded the 80% benchmark, demonstrating high compression ratios for lossless images. From the original 17.63 GB of the significant medical dataset, the total compression achieved was 8.62 GB, which is very efficient for medical repository storage. These proven results give a clear insight into how well the hybridization and fusion of these techniques have yielded results and give us a significant boost and inspiration to use this well-constructed technology.

### 4.4. Compression Analysis

The process of image steganography, transformation, and compression can be performed on any platform using any method. However, it is essential to have higher security and minimal storage space. Therefore, choosing an efficient and effective method will give the best results. Choosing a fast KT optimized for steganography and DWTs, along with wavelet packet compression of images, gives better results. To demonstrate the success of the proposed method, experiments were performed in MATLAB version R2017b, in which the data embedding, transformation, and compression work was carried out. MATLAB software is a tool that can be used to perform all mathematical operations, providing highly accurate results at every step. Windows 10 OS provides excellent support for productive and skillful research. KT lends an innovative flavor to steganography techniques, and transformations help protect steganography images. Additionally, the transformed image is compressed using lossless compression and decompressed again at the receiving end to the exact same resolution as the original image. The compressed images are stored in the database in less space, which helps in efficient storage.

### 4.5. Comparative Results and Statistical Analysis

The performance of the proposed method was also evaluated for various JPEG and PNG image formats to obtain compression ratios with percentages of compression. The image sizes considered for the simulation were 128 × 128, 256 × 256, 512 × 512, and 1024 × 1024. The performance measures of the proposed model are shown in Table 2. The expected performance in terms of compression ratios to other models is detailed in Table 3. The expected performance in terms of compression percentages to other models is shown in Table 4. The expected performance in terms of computing times for the proposed model to other models is shown in Table 5. The expected compression and decompression computing times for the benchmark datasets for the proposed model in relation to those of the other models are shown in Table 6 and Table 7, respectively.

The following significant inferences were drawn based on the simulation.
▪JPEG formats’ compression ratio percentages were slightly higher than PNG formats. When the image size increases, that is, for high-resolution images, the compression ratio lies between 7% and 7.5%, and the compression percentage lies between 30% and 37%.▪The proposed model increases the expected compression ratio and percentage compared to other models. The average compression ratio lies between 7.8% and 8.6%, and the expected compression ratio lies between 35% and 60%.▪Computing time is reduced with the proposed method relative to other methods. For high-resolution images, the expected computing time lies between 4 ms and 5 ms compared to other approaches.

The limitations of the proposed model are as follows:▪The memory requirement through DWT is higher since it processes the complete image.▪Training and inference can be achieved by designing an optimal neural network to reduce complexity.

## 5. Conclusions and Future Work

In this research, a picture of a brain hemorrhage was taken and entered into the patient’s records. The input image was transformed using a DWT “branch” wavelet. The transformed image was then compressed using “Haar” wavelet compression. The image was compressed to 83.3333% and decompressed and reconstructed to obtain the embedded brace image. This downloaded brace image was then used to extract patient information from the original image. After decompression, the resolution of the studied image was never disturbed, and since the brace image was lossless, the extracted patient data were unchanged. The optimization obtained from the test significantly contributed to the preservation and long-term archiving of the medical image without compromising the patient’s privacy. The compression technology produced an amazing lossless image after decompression, which was useful for retrieving patient information. The method developed in this research ensured that medical images were archived optimally and increased privacy. JPEG formats’ compression ratio percentages were slightly higher than those of PNG formats. When image size increases, that is, for high-resolution images, the compression ratio lies between 7% and 7.5%, and the compression percentage lies between 30% and 37%. The proposed model increases the expected compression ratio and percentage compared to other models. The average compression ratio lies between 7.8% and 8.6%, and the expected compression ratio lies between 35% and 60%. Computing time is reduced with the proposed method relative to other methods. For high-resolution images, the expected computing time lies between 4 ms and 5 ms.

In the future, the Internet of Medical Things, medical imaging strategies, soft computing with evolutionary operators, and other hybrid image processing strategies will be applied further to reduce the complexity of the proposed model and to develop better recommender systems for large-scale, high-quality medical images.

## Figures and Tables

**Figure 1 bioengineering-10-00333-f001:**
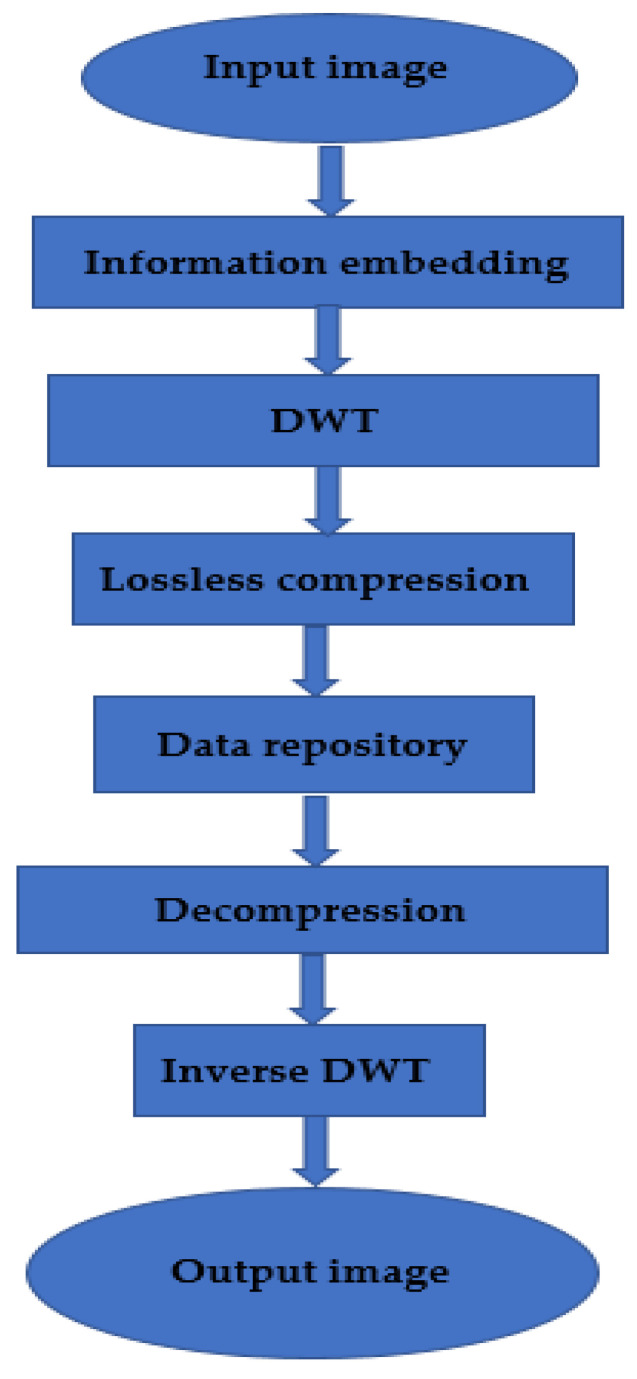
Overview of the architecture.

**Figure 2 bioengineering-10-00333-f002:**
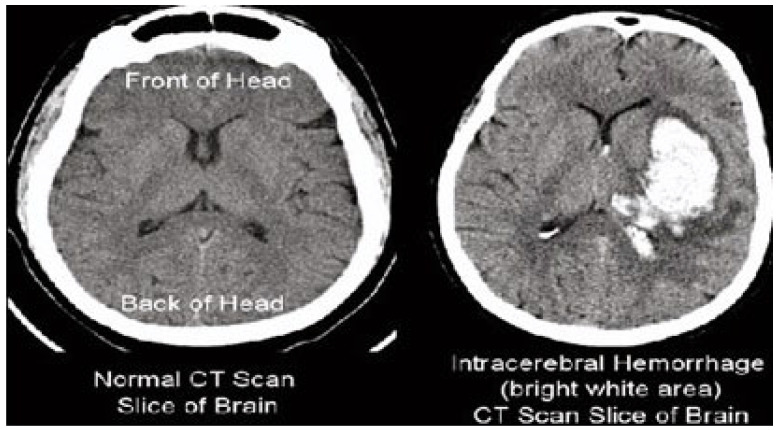
The embedded image.

**Figure 3 bioengineering-10-00333-f003:**
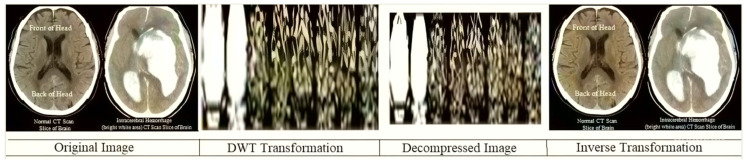
Hybrid transformation and compression.

**Figure 4 bioengineering-10-00333-f004:**
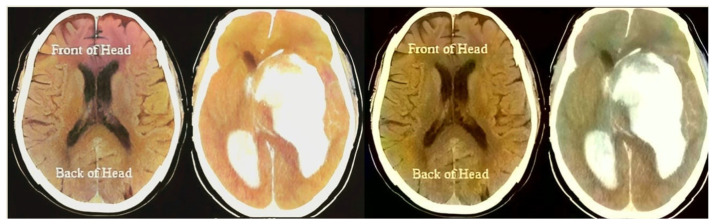
Applying DWT.

**Figure 5 bioengineering-10-00333-f005:**
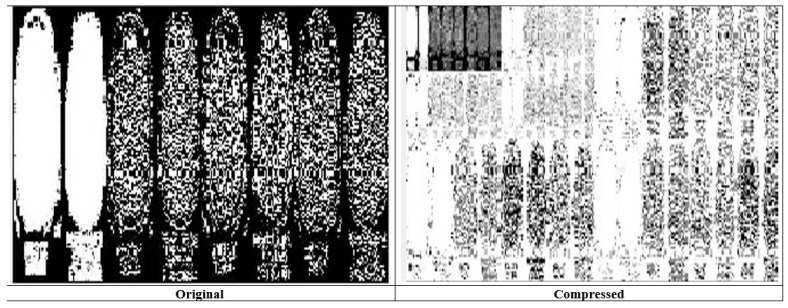
Histograms of original and compressed images.

**Figure 6 bioengineering-10-00333-f006:**
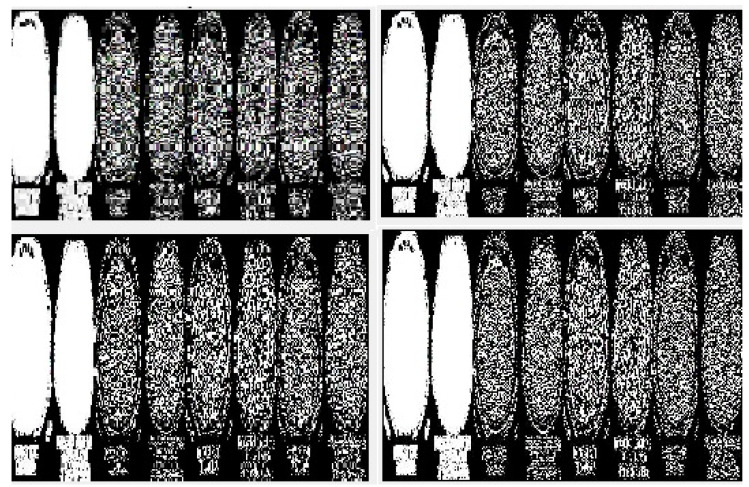
Compression of the transformed image.

**Figure 7 bioengineering-10-00333-f007:**
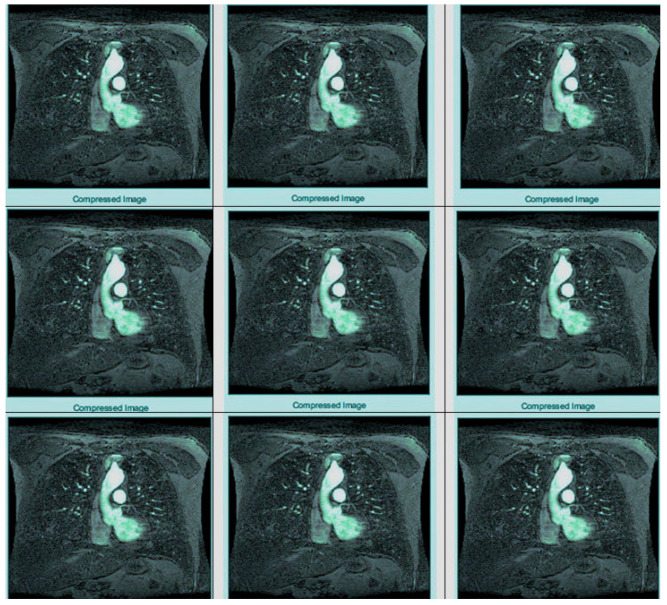
Multiple compression of MRI using different wavelet families.

**Figure 8 bioengineering-10-00333-f008:**
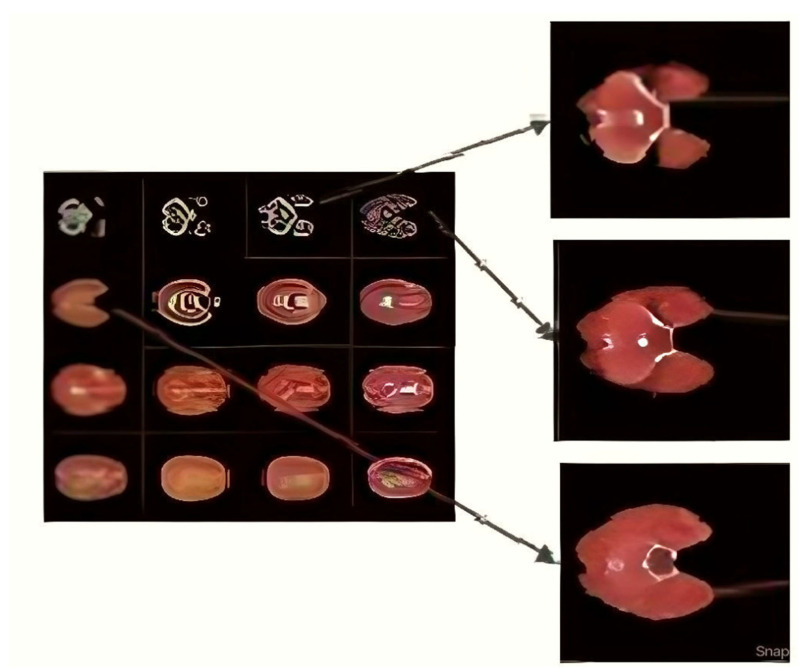
MRI compression using core wavelet families.

**Table 1 bioengineering-10-00333-t001:** Comparative analysis.

Measures	Original Image	Decompressed Image 1	Decompressed Image 2
MSE	41.89	41.88	41.88
SNR	25.79	25.79	25.79
PSNR	31.92	31.91	31.91
SSIM	0.99	0.99	0.99
BER (number, ratio)	1548, 7.38%	1548, 7.37%	1548, 7.37%

**Table 2 bioengineering-10-00333-t002:** JPEG and PNG format comparisons.

Size	Features	JPEG	PNG
128 × 128 Image 1	Compression ratio	8.9%	8.7%
128 × 128 Image 1	Compression %	52.5%	46.6%
128 × 128 Image 2	Compression ratio	8.5%	8.2%
128 × 128 Image 2	Compression %	53.5%	41.5%
256 × 256 Image 1	Compression ratio	8.1%	8.0%
256 × 256 Image 1	Compression %	42.5%	41.2%
256 × 256 Image 2	Compression ratio	7.5%	7.2%
256 × 256 Image 2	Compression %	36.2%	33.5%

**Table 3 bioengineering-10-00333-t003:** Expected performance—compression ratios compared with those of other models.

Size	Proposed Model	Other Models
128 × 128	8.6%	8.1–8.5%
256 × 256	8.4%	8.1–8.3%
512 × 512	7.9%	7.5–7.8%
1024 × 1024	7.8%	7.7–7.8%
Public datasets	8.5%	7.8–8.4%

**Table 4 bioengineering-10-00333-t004:** Expected performance—compression percentages compared with those of other models.

Size	Proposed Model	Other Models
128 × 128	58.6%	58.1–58.3%
256 × 256	48.4%	48.1–48.3%
512 × 512	57.2%	47.5–47.9%
1024 × 1024	37.8%	35.7–36.4%
Public datasets	58.4%	45.5–55.8%

**Table 5 bioengineering-10-00333-t005:** Expected performance—computing times (in ms) compared with those of other models.

Size	Proposed Model	Other Models
128 × 128	3.5	5.7
256 × 256	3.8	6.2
512 × 512	4.2	6.4
1024 × 1024	4.7	7.5
Public datasets	35.2	48.4

**Table 6 bioengineering-10-00333-t006:** Expected performance—compression computing times (in ms) compared with those of other models.

Size	Proposed Model	Other Models
128 × 128	2.7	3.8
256 × 256	3.2	5.9
512 × 512	4.1	5.4
1024 × 1024	4.2	5.5
Public datasets	12.56	21.78

**Table 7 bioengineering-10-00333-t007:** Expected performance—decompression computing times (in ms) compared with those of other models.

Size	Proposed Model	Other Models
128 × 128	3.8	4.7
256 × 256	4.3	5.2
512 × 512	4.3	6.2
1024 × 1024	4.5	7.8
Public datasets	15.23	25.35

## Data Availability

Not available.

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
