# Peer review of "Modelling and Analysis of Hybrid Transformation for Lossless Big Medical Image Compression"

_bioengineering, 2023, doi:10.3390/bioengineering10030333_

Round 1
Reviewer 1 Report
The manuscript demonstrates an application of image compression for the purpose of storing medical data. The manuscript is well written and scientifically sound, but I believe it is not in-depth enough as it currently stands. My comments follow below:
1. If I understand correctly, only a single image (CT of a brain hemorrhage) is tested. This is not enough to lay any claims regarding the performance. Authors should test on multiple images and present a statistical analysis of the results, in order to truly demonstrate the performance of their algorithm. This can be done using the publicly available dataset the authors state they have used (but do not present the results from testing beyond the compression level).
2. "BuddhaSmiledWholeHeartedly", "BuddhaSmiled", "Bud", SSIM, BER are therms used in the Table 1, presenting main results, that are not explained anywhere in the rest of the manuscript. I don't believe that these terms are so well known that they can be used without introduction. Authors should add these explanations
3. There are a lot of similar errors. For example, in some places authors use JPF and in some JPG. Which format was actually used? Considering the focus on the image compression this cannot be unclear.
4. One of the common issues of the compression or encryption use on data is the time necessary for decompression. Authors should measure and comment, in depth, on the time it takes for the proposed method to compress and decompress images on a larger scale than a single image. This can also be performed on the public dataset Authors mention in 4.1.
5. in "4.3. Comparative Results & Analysis" - what data is the comparison performed on? Which other methods are used for comparison?
6. There is a number of poorly written and hard to understand sentences. I would suggest a re-check of the manuscript text by the authors.
Author Response
Response to reviewer 1 is attached.

Reviewer 2 Report
This research paper presents a hybrid approach to secure and optimize the storage of large medical data in medical data repositories. The approach uses advanced steganography, wavelet transform (WT), and lossless compression. Here are some suggestions to the authors:
1. The authors need to highlight the motivation behind this work and justify the choices of all processes along the pipeline
2. The authors need to update the literature review and related work especially the ones that came up with similar systems addressing the similar issue
3. The authors need to highlight the shortcomings and limitations and provide future directions
4. The authors need to significantly enhance the quality of the images and show some examples High quality colored images if possible
Author Response
Response to reviewer 2 is attached.

Round 2
Reviewer 1 Report
The authors have addressed al of the comments that were given to them. I believe the paper is now ready for publication.
Reviewer 2 Report
the authors addressed the comments